# Neutron Activation Analysis of Rare Earth Element Extraction from Solution through a Surfactant-Assisted Dispersion of Carbon Nanotubes

**DOI:** 10.3390/nano14010092

**Published:** 2023-12-28

**Authors:** Adam Samia, Donald Nolting, Joseph Lapka, William Charlton

**Affiliations:** School of Nuclear and Radiation Engineering, University of Texas, Austin, TX 78758-445, USA; don.nolting@austin.utexas.edu (D.N.); joseph.lapka@austin.utexas.edu (J.L.)

**Keywords:** carbon nanotubes, dispersion, extraction, graphene oxide, gum arabic, neutron activation analysis, rare earth elements, scanning electron microscopy, surfactant, Triton X-100

## Abstract

We report the preparation of surfactant-assisted carbon nanotube dispersions using gum arabic, Triton X-100, and graphene oxide as dispersing agents for removing rare earth elements in an aqueous solution. The analytical tools, including (a) scanning electron microscopy and (b) neutron activation analysis, were utilized for qualitative and quantitative examinations, respectively. Neutron activation analysis was employed to quantitatively determine the percent of extraction of nuclides onto the carbon structure, while the images produced from the scanning electron microscope allowed the morphological structure of the surfactant–CNT complex to be analyzed. This report tested the effects responsible for nuclide removal onto CNTs, including the adsorbent to target mass ratio, the CNT concentration and manufacturing process, the pH, and the ionic radius. Observable trends in nuclide extraction were found for each parameter change, with the degree of dispersion displaying high dependency.

## 1. Introduction

Continual increases in industrial processes have resulted in the contamination of neat water sources with various rare earth elements (REEs) [1]. Not only have these metals been found to have detrimental effects on the central nervous system [2], but their unique physicochemical, optical, and magnetic properties make their use pivotal in various technological advancements [3]. The heightened demand for these metals leads to the increased production of REE-containing ores, which ensures increasing exposure to the public. Hence, it has become crucial to remove these metal ion contaminants from various effluent sources to ensure no contamination levels higher than those acceptable to the public and to recover these scarce metals for a steady and regular industrial supply. Although an assortment of adsorbents has been investigated [4,5,6], nano-based materials are becoming increasingly sought after due to their large surface area and numerous active adsorption sites [7].

Carbon nanotubes (CNTs) and single-, double-, and multi-walled cylindrical graphene sheets were first synthesized by Lijima in 1991 [8]. Since this discovery, these sp^2^-bonded graphitic structures have been extensively investigated due to their unique characteristics (i.e., a sizeable surface area to volume ratio, hollow nano-sized tubes, and various electronic, mechanical, and chemical properties) [9,10]. Unfortunately, using CNTs in industrial applications is hindered by their extremely poor solubility in both aqueous and organic media [11]. In solution, CNTs tend to aggregate in what is known as a “bundled” state. Due to an extensive network of van der Waals interactions between adjacent nanotubes, the tendency for a bundle to form originates from a strong tube/tube cohesion [12]. For example, a pair of (10, 10) tubes (in which the chiral indices of the form (n, m) specify the perimeter of the carbon nanotube on the graphene net [13]) resulted in cohesive energy of −40 kT nm^−1^ [14]. Considering that nanotubes can have lengths of up to tens of microns, this cohesive energy can become rather significant and result in very stable bundles. Such high interaction energy makes CNT dispersion a cumbersome and challenging task.

For most applications, CNTs should be pretreated to form a highly dispersed colloid. Both physical and chemical methods ameliorate the dispersion of CNTs [15]. Physically, ultrasonication can disperse nanotubes in an aqueous media. The high-energy ultrasound waves can structurally damage the nanotubes, thus resulting in lowered cohesive energies and less bundling [16]. However, from the same observation, prolonged sonication times can increase the disorder of the CNT structure and ultimately lead to the formation of amorphous carbon [16]. Chemically, CNT bundling is alleviated through covalent or non-covalent methods. Covalent functionalization of the CNT sidewalls with various chemical moieties can improve dispersion; however, this consequently alters the properties of the CNTs, which can often be undesirable in many applications [17]. In the non-covalent approach, the chemical moiety (usually an amphiphilic surfactant) is adsorbed onto the nanotube surface either through π–π stacking for an uncharged species or surface coulomb interactions for charged species [18]. Depending on the application, one or more of these dispersion techniques can obtain a highly suspended CNT colloid.

Because the non-covalent approach preserves the electrical properties of the carbon structure, the present study aims at comparing the extractability of nuclides in solution with as-received CNTs dispersed with various surfactant materials (gum arabic (GA), Triton X-100, and graphene oxide (GO)). The chemical structure of each surfactant is provided in Figure 1a–c.

Each surfactant tested displays varying degrees of dispersibility, which can vastly alter the CNTs’ adsorptive capabilities. Gum arabic is a highly branched, natural product composed of a complex mixture of macromolecules [19]. It has been reported that the homogenous dispersion of CNTs in a GA solution disrupts cohesive bundling in the crystalline ropes [20]. Gum arabic shows favorable adsorbability to the CNT surface by utilizing a two-phase interface comprising a horizontal layer of small thickness forming a steric hindrance coating followed by surface aggregation by strong hydrophobic interactions. CNTs lie in the boundary between two directional planes, which reduces the interfacial tension and improves the hydrophilicity and dispersibility of the nanotubes [21]. Triton X-100, alkylphenol hydroxypolyethylene, is an amphiphilic molecule with a hydrophobic head and hydrophilic tail that creates a micellar formation around the nanotubes. The alkylphenol head group is attracted to the CNT surface via π–π stacking, while the polyethylene chain increases the spatial volume and steric hindrance, thus providing greater repulsive forces between individual nanotubes [22]. Then, graphene oxide, a near two-dimensional macromolecule with multiple aromatic regions and a wide array of hydrophilic oxygen groups (carboxylic acids, hydroxyls, carbonyls, and epoxy groups), has also been shown to improve dispersibility [23]. The π-conjugated central plane of the GO sheet facilitates interaction with the CNT sidewall, while the oxygen-containing functional groups help to maintain the water stability of the CNT–GO complex [24].

The dispersibility of CNTs and extractability of REEs in a solution of each surfactant have been analyzed experimentally and qualitatively based on the structural organization of the CNT–surfactant complex. This report quantitatively describes the extraction of REEs from solution using CNTs as the adsorbent via neutron activation analysis (NAA). Although other analytical tools can be used to quantify REE extraction, NAA was chosen for its sensitivity towards trace metals in highly dense and opaque carbonaceous sample mixtures with minimal sample processing. This technique is also readily available for a minimal cost at UT-Austin. The percent adsorption on the CNTs was calculated by determining the ratio of activities in each aliquot produced through experimentation. Scanning electron microscopy (SEM) was employed to analyze the dispersibility of these surfactants, such that comparisons can be drawn from the images captured. This study evaluated the maximum extractability of REEs in solution by altering the pH, the concentration of the surfactants, and the CNT concentration and experimenting with different CNT manufacturing methods. The results are presented herein.

## 2. Materials and Methods

### 2.1. Materials

All reagents were purchased from commercial sources and used without further purification unless noted otherwise. The solutions in these experiments were all prepared with 18.2 MΩ cm (Thermo Scientific Water Purification System, Waltham, MA, USA) water at room temperature, 21 ± 1 °C. Ultrasonication was performed with a Branson CPX2800 Ultrasonic Bath (Emerson Electric Co, Clayton, MO, USA) operated at 40 kHz.

Single-walled CNTs (SWCNTs) were procured from two different vendors: Sigma-Aldrich (SA-CNT) (≥95% carbon basis; 0.84 nm average diameter; 1 μm average length) and Cheap Tubes (CT-CNT), which are labeled as single-walled–double-walled (≥95% carbon basis; 0.8–1.6 nm diameter; 5–30 μm length). A comparison of each CNT is given in Table 1.

The surfactants (gum arabic (Sigma-Aldrich, St. Louis, MO, USA), Triton X-100 (Alfa Aesar, Haverhill, MA, USA), and graphene oxide (Cheap Tubes, Grafton, VT, USA)) were used as received. Nitric acid (Sigma-Aldrich) and ammonium hydroxide (Sigma-Aldrich) were used to adjust the surfactant pH values and were used as received.

The target nuclides Sc, Y, La, Tb, and Yb were all purchased from Sigma-Aldrich in their hydrated nitrate forms with greater than 99.9% trace metal bases. They were used as received.

Samples were analyzed using an ORTEC or Canberra High Purity Germanium (HPGe) Radiation Detector with Genie 2000 analysis software (V3.4.1).

### 2.2. Reactor Facilities

Neutron activation analysis was conducted using the Nuclear Engineering Teaching Laboratory (NETL) Mark II TRIGA (Training, Research, Isotope, General Atomics) reactor at the University of Texas in Austin. This 1.1 MW reactor was operated at a power level of 900 kW for 30 min with samples in the rotary specimen rack (RSR). This irradiation facility has an approximate flux equal to 2 × 10^12^ n cm^−2^ s^−1^, which provides sufficient fluence to the target samples.

### 2.3. Preparation and Separation of CNT Dispersion

Dispersions were produced for varying experimental setups to compare the dispersibility of each of the three surfactants. CNTs (at concentrations of 1, 3, 5, or 10 mg/mL) were added to the surfactant (pH-adjusted) and sonicated for 1 hr. The surfactant concentrations used in this study were found from the literature and are as follows: 15 *w*/*w*% GA [20], 8% Triton X-100 [18], and 1 mg/mL of GO [25]. These surfactant concentrations were also tested experimentally, as shown in Appendix A, to determine the optimal dispersibility and viscosity for adsorption. Through experimentation, the surfactant concentrations from the literature were found to display ideal characteristics and were therefore used throughout this study. The “no surfactant” experiment utilized 0.2 mL of total solution (water and target solution) to wet the CNTs. The pre-irradiated target solution was then added to the dispersion at varying CNTs to target mass ratios and re-sonicated to facilitate homogenization. The emulsion was left overnight to ensure equilibrium had been reached. The following morning, the samples were centrifuged, and the supernatant target solution was decanted. Water was added to the CNTs, and this process of centrifugation and decantation was repeated to obtain three wash aliquots. All aliquots (CNTs, target solution, and three wash solutions) were then analyzed using an HPGe detector. For each experiment set (CNT to target ratio, CNT concentration and manufacturer, and pH adjustments), a Yb target was used. The top result, or the result with the highest percent of extraction, was then replicated with each of the five target nuclides listed.

### 2.4. SEM Characterization and Analysis

The dispersion of a nanotube suspension was characterized using a scanning electron microscope (Hitachi S-5500, Tokyo, Japan) operating at 30 kV and a current of 15 μA. At an unadjusted pH, approximately 3–5 drops (~50 μL per drop) of the 3 mg/mL CNT loading suspensions were placed directly onto an aluminum SEM sample holder (Rectangular Specimen Mount—Ted Pella, Inc., Redding, CA, USA) and allowed to dry overnight. Before analysis, each sample was coated with platinum using an EMS (Electron Microscopy Sciences, Hatfield, PA, USA) Sputter Coater operated at 40 mA with a 9 × 10^−2^ mbar vacuum applied.

### 2.5. Determination of REE Extraction Using Neutron Activation Analysis

Neutron activation analysis can characterize the dispersibility of CNTs and the subsequent extractability of nuclides in a solution. To illustrate the extractability of CNT–surfactant dispersions using NAA, activity values were recorded for each aliquot per experiment. The photopeak for each nuclide was chosen so that interference from competing branching ratios was minimized, and they are as follows: scandium (^46^Sc—889.28 keV), yttrium (^90^Y—202.51 keV), lanthanum (^140^La—487.02 keV), terbium (^160^Tb—879.38 keV), and ytterbium (^175^Yb—396.33 keV). The percent extractability Equation (1) of the desired nuclide can then be calculated as:(1)$$\%\;Extractability=A_{CNT}/\sum_{i}A_{i}\times 100\%$$
where *A_CNT_* is the activity value of the CNT aliquot and *A_i_* is the activity value for each *i*th-constituent (target, CNT, wash 1, wash 2, and wash 3).

## 3. Results and Discussion

### 3.1. CNT Dispersion Characterization through SEM

Direct evidence for the suspension of CNTs by surfactant materials can be supported through SEM observation. SEM micrographs of each surfactant–CNT dispersion at 100× and 50,000× magnification have been qualitatively analyzed and are given in Figure 2c–h. For comparison, an SEM image of pristine SA-CNTs, nanotubes that have not undergone surfactant-aided dispersion, has also been included in Figure 2a,b. The image provided in Figure 2e,f of CNTs dispersed with Triton X-100 was difficult to acquire due to the inability of Triton X-100 to fully dry onto the sample mount, and it was thus imaged as a sludge material.

The morphology of the non-surfactant-suspended CNTs shows severe agglomeration in solution. Without a surfactant, the CNTs are present as bundles or ropes. The addition of a surfactant into the solution exfoliates the CNTs and allows for successful dispersion. This unraveling of the CNT bundles can be seen when comparing the 100× to 50,000× SEM magnifications. Based on the images provided in Figure 2c–h, one can conclude that the dispersion properties of each surfactant greatly vary. For GA, the adsorption onto the CNTs disrupts the van der Waals attractions among the aggregated CNTs. This leads to a repulsive intertube potential that results in a thermodynamically stable nanotube dispersion in the GA crystalline structure [26]. For Triton X-100, although the image was unable to be appropriately focused, there is a distinct observational analysis compared to the non-surfactant-suspended CNT image. CNT dispersion using the non-ionic Triton X-100 surfactant is facilitated by the hydrophilic groups of the surfactant forming a solvation shell around the nanotubes [27]. The π–π interaction between the benzene ring and the CNT surface is maximized as the hydrophobic polyoxyethylene chain provides repulsive forces between individual nanotubes, thus allowing for successful dispersion [28]. Furthermore, as it can be seen from the SEM image of the GO sample, small, hair-like CNTs were adsorbed onto the GO surface. Due to the presence of multiple π-conjugated aromatic domains within the basal plane, the GO surfactant was found to form a water-suspended GO–CNT complex after disentanglement of the CNT bundles and interaction with individual nanotubes [29].

### 3.2. Nuclide Extraction Percentage for Increasing CNT to Target Mass Ratio

The dependence of the SA-CNT to target mass ratio on nuclide extractability is shown in Figure 3a–c. Regardless of the surfactant used, the adsorption and extractability of nuclides in solution at low CNT to target mass ratios result in minimal uptake onto the nanotube adsorbents. However, increasing this ratio promoted a steady increase in extraction until a steady-state value was obtained above a 20 CNT to 1 Yb mass ratio. An increase in adsorbent to target mass ratio from 1 CNT:1 Yb to 30 CNT:1 Yb resulted in heightened adsorption from 15.6% to 62.8%, 7.3% to 82.1%, 13.1% to 93.2%, and 7.0% to 73.1% for GA, Triton X-100, GO, and no surfactant, respectively. As the amount of adsorbent relative to the target increases in solution, there is an increased CNT surface area accessible to the nuclides. A higher surface area of the nanotubes results in more active contact sites available for adsorption. Nuclides can then be adsorbed onto the carbon nanostructure and removed from the solution in high percentages.

At the highest mass ratio tested, GA had a lower capture percentage at 62.8% than no surfactant, Triton X-100, and GO at 73.1%, 82.1%, and 93.2%, respectively. This analysis can be explained with a greater understanding of the structure and morphology of GA in solution. The gelatinous nature of GA hinders the mobility of the nuclides in the solution. It has been postulated that this decrease in mobility disallows a portion of the nuclides to interact with the CNT surface capture sites. Gum arabic also exhibits a highly branched array of carbohydrates and proteins. This chemical composition, consisting of a vast amount of oxygen-containing functional groups, shows a favorable affinity towards cationic ions (i.e., Yb^3+^) in solution, which results in the capture of the desired nuclide in the surfactant material rather than the expected CNT structure.

Both Triton X-100 and GO as surfactants resulted in an increase in capture at the highest mass ratio tested compared to the no surfactant experiment. The inclusion of Triton X-100 and GO as surfactant materials in the experimental protocol has demonstrated an enhancement in nuclide extraction onto carbon nanotubes compared to experiments conducted without surfactants. Triton X-100, a nonionic surfactant, and GO, a two-dimensional macromolecule, were selected for their ability to promote ideal dispersion in aqueous solutions. The surfactants create a stable environment by reducing the interfacial tension between the CNTs and the surrounding solution. This enhanced dispersion facilitates increased contact between the CNTs and nuclides, thus providing more active sites for adsorption. Additionally, when exploring the use of GO as a surfactant, the heightened capture is likely due to the low viscosity of the GO surfactant compared to Triton X-100. This lowered viscosity resulted in a higher mobility and, subsequently, a higher capture of the Yb species.

### 3.3. Extraction with Varying CNT Concentrations from Both Suppliers

After it was shown that a majority of the nuclides in the solution could be adsorbed onto the CNT structure at higher CNT to Yb mass ratios, a series of experiments were conducted to determine the optimal CNT concentration. The results are shown in Figure 4a–f. Each experiment was performed at a 20 CNT:1 Yb mass ratio with CNT concentrations of 1, 3, 5, or 10 mg/mL for each CNT process of manufacturing. All samples were dispersed in 3 mL of the selected surfactant.

As the name implies, Cheap Tubes are manufactured using a process most recognized for its scalability at viable production costs [30]. The floating catalyst CVD synthesis route allows for continuous CNT production, with the catalyst and the hydrocarbon being introduced into the reaction furnace simultaneously [31]. This continuous production scheme results in CNTs that consist of a combination of single-walled and double-walled CNTs [32]. The other CNT manufacturing technique tested, Sigma-Aldrich CNTs, are synthesized using a patented Co-Mo catalyzed CVD technique [33]. This process utilizes a non-metallic cobalt species that is stabilized with molybdenum oxide (MoO_3_) before being reduced by a carbon compound, carbon monoxide (CO), in the reaction chamber [34]. This approach results in high selectivity towards CNTs of very small diameters and a narrow distribution of CNT structures (i.e., a high percentage of SWCNT structures); however, the purification process results in carboxylic acid functionalization of the nanotubes [34]. Figure 4 shows that the overall trend for each surfactant remains consistent for the two CNT manufacturing processes tested, Co-Mo catalyzed CVD for SA-CNTs and floating catalyst CVD for CT-CNTs. However, when comparing the absolute value of adsorption, there is significant variance in the percent of extraction.

When comparing the two CNT manufacturing methods, it becomes clear that the higher-quality and more expensive Sigma-Aldrich CNTs perform better at removing nuclides from the solution. Multiple explanations for this phenomenon have been hypothesized, and the higher performance is most likely caused by a combination of two or more. As it turns out, the carboxylic acid functionalization of the SA-CNTs plays a crucial role in nanotube dispersion and nuclide extraction. Oxidized carbon nanotubes have a significant increase in hydrophilic properties compared to non-functionalized CNTs. Oxygen-containing moieties on the CNT surface decrease the highly prohibitive bundling energy barrier caused by van der Waals forces of adjacent nanotubes [35]. This, along with the slightly negative charge associated with oxygen groups, allows for efficient capture of the cationic species, such as the REEs. The SA-CNTs also exhibit an average length far less than the provided CT counterpart. Sigma-Aldrich CNTs were reported as having an average tube length of 1 μm compared to the Cheap Tubes CNTs, which have a range of lengths from 5 to 30 μm. Considering that the cohesive bundling energy is inversely related to nanotube length, the CT-CNTs, with lengths 5 to 30 times longer than SA-CNTs, are energetically more prone to exist in the bundled state and are thus more challenging to use to achieve effective dispersions. The purity of the CNT content from each manufacturing method has also been evaluated as a potential explanation for the decrease in nuclide capture from SA to CT-CNTs. The CT-CNTs are produced in a manner that results in a combination of single-walled and double-walled CNTs. For an equal mass loading of Sigma-Aldrich vs. Cheap Tubes CNTs, there are more individual strands of the SA-CNTs. This causes the CT-CNT samples to be under-dispersed in the optimal surfactant to CNT mass loadings. At the same time, due to the higher surface area of the SA-CNTs in solution, there are also more vacant capture sites, resulting in the observed higher capture percentages. This combination of single-walled and double-walled CNTs resulting in lower capture percentages was further investigated with multi-walled CNTs (MWCNTs), as shown in Appendix A. MWCNTs consist of nanotubes with multiple (more than two) concentric graphene cylinders [36], and they follow the trend of decreased adsorption compared to SWCNTs for equal mass loadings.

Regardless of the CNT manufacturing process, a trend emerges among the three surfactants tested. At low CNT concentrations, the system is over-dispersed. That is, a sufficient amount of surfactant is available to coat the nanotube surface evenly. Eventually, a CNT concentration is reached where the surfactant loading is optimized. This is where the maximum extractability is obtained (>10 mg/mL, <1 mg/mL, and 3 mg/mL for GA, Triton X-100, and GO, respectively). For subsequent increases in CNT concentration, insufficient surfactant is available to suspend the CNTs fully, and the nuclide extraction will decrease in the now-under-dispersed system. Due to the physical limitations of centrifuging and separating CNT solutions with concentrations greater than 10 mg/mL, these concentrations were not evaluated.

This idea was explored in detail when Rastogi et al. reported a comparative analysis of the dispersion of MWCNTs with varying surfactants [18]. Their study depicted a linear increase in adsorption with an increase in the concentration of nanotubes. At this point, the maximum extraction was achieved, and further increases in CNT concentrations resulted in lower adsorption percentages.

### 3.4. Extraction Percentage with Varying pH

The pH of the reaction solution is one matrix property that has been observed to have a profound effect on the adsorption of REEs. The influence of pH on nuclide adsorption can be explained by considering the impact this parameter has on the surface properties of the CNTs and the hydrolysis capacity of the target [37]. In a publication by Tan et al. investigating the pH dependency of Eu and Am adsorption on oxidized MWCNTs [38], they reported a similar trend to this study. In their analysis, the adsorption of nuclides onto nanotubes displayed a clear pH dependency, which suggested a heightened interaction with surface active sites, like hydroxyls and carboxylate groups, as they became progressively deprotonated. The large concentration of deprotonated, oxygen-containing functional groups offered a more hydrophilic structure and an increase in the ion-exchange capability of the nanotubes.

As seen from Figure 5a–c, the oxygen-containing moieties attached to either the CNT surface or the surfactant material can alter the ion-exchange capability of the carbonaceous material [38].

Protic oxygen-containing functional groups undergo protonation or deprotonation reactions depending on the solution pH [39]. The low extraction percentages in acidic media can be attributed to the interference and competition between hydronium and the metal cation for the same active capture sites [37]. From the results obtained, the highly mobile hydronium ions were found to experience a higher affinity towards these capture sites compared to the metal cation, and they render the CNT surface positively charged. Thus, at low pH, this protonation promotes the formation of water molecules on the CNT surface and weakens the hydrogen bonds between the CNTs and the target material, thus hindering adsorption [40]. As functional groups deprotonate with increasing pH, the adsorption improves on the now-ionized nucleophilic sites. When pH goes up, the concentration of hydronium ions in the solution decreases, which alters the net surface charge to negative. The polarized oxygen-containing functional groups can then electrostatically attract the metal cation and form a metal–ligand complex [37]. This has been found to be the main driving force of nuclide adsorption onto carbon nanotube structures [41]. In highly basic reaction solutions with pH values above nine, the target nuclide begins to precipitate due to more intense hydrolyzation [42]. Nevertheless, the observed increases in adsorption from pH changes from two to nine reveal that electron donor effects between the CNTs and the nuclides of interest are highly pH-dependent.

### 3.5. Determination of REE Extraction Percentage

After optimization amongst the CNT to adsorbent ratios, CNT concentrations and manufacturers, and pH adjustments, the most optimized experiment was tested for use with various REE target nuclides. The REE target nuclides tested include Sc, Yb, Y, Tb, and La, which have been ordered by their six-coordinate ionic radii, 88.5, 100.8, 104.0, 106.3, and 117.2 pm, respectively. Each experiment was conducted using a 20 SA-CNT to 1 target mass ratio at a CNT loading of 3 mg/mL in a pH 9.37 solution of 1 mg/mL of GO. As seen in Figure 6, the percent of nuclides that can be extracted from the solution depends on the ionic radii of the desired nuclide.

The structure and size distribution of CNT pores between bundles and individual tubes are rather complex, and they have been found to exist in a wide range of sizes (<5–100 nm) [43]. Thus, it has been inferred that the adsorption of REEs is based on the ionic radii of each nuclide and the degree to which the CNTs are functionalized. The CNTs, which are rigid, rod-like structures in solution, are functionalized with carboxylic acid groups (the scale to which they are functionalized was not provided by the manufacturer). These oxygen-containing moieties increase the electron charge density of the CNT structure, which allows for efficient capture of the cationic species. However, the affinity of the cationic ion towards the electron-rich ligand depends on the ionic radii of the target nuclide. For a nuclide with a smaller ionic radius, i.e., Sc, the adsorption of the nuclide is limited by the number of carboxylic acid groups it can interact with. As such, it requires more energy to displace a water molecule from the CNT hydration sphere than it does for an equivalently charged ion of a larger size [44]. For a nuclide with a larger ionic radius, the interaction with more carboxylic acid groups creates an energetic advantage for adsorption onto the sorbent compared to smaller ions. Correspondingly, the sorptive affinity for ions follows a common periodic table trend based on ionic radii within a group.

## 4. Conclusions

Carbon nanotubes continue to be increasingly pursued as a sorbent material. Their significant adsorption capacity can be used as a promising tool for the removal of REE nuclides. Herein, the dispersion of CNTs and the extractability of REEs in solution were evaluated experimentally and qualitatively for three different surfactant materials: gum arabic, Triton X-100, and graphene oxide. The dominant theme of this report was to analyze parameter changes responsible for the suspension of CNTs in a surfactant medium. Each parameter tested, including the adsorbent to target mass ratio, the CNT concentration and manufacturer, the pH, and the ionic radius, resulted in observable trends in nuclide extraction. These trends were highly dependent upon the CNT degree of dispersion. The adsorption and extraction of nuclides onto the CNT structure were low for both under- and over-dispersed samples. As the system became more optimally dispersed, nuclides in solution were readily sorbed onto the CNT structure. From the results obtained in this study, it can be concluded that for the efficient capture and removal of nuclides in the solution (i.e., >90% extraction), an unadjusted to slightly basic pH solution of Triton X-100 or graphene oxide should be used as the surfactant medium at CNT to target mass ratios higher than 20 to 1 with 1 mg/mL or 3 mg/mL CNT loadings for Triton X-100 or graphene oxide, respectively. The adsorbed nuclide can then be quantitatively removed and recovered from the carbonaceous structure by mixing with a dilute acid, as shown in Appendix A. In summary, we have found that the surfactants gum arabic, Triton X-100, and graphene oxide can effectively disperse CNTs, thus allowing for successful capture and removal of REE nuclides in solution.

## Figures and Tables

**Figure 1 nanomaterials-14-00092-f001:**
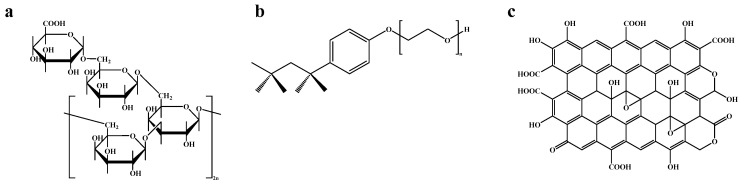
Chemical structure of each surfactant ((**a**) gum arabic; (**b**) Triton X-100; and (**c**) graphene oxide).

**Figure 2 nanomaterials-14-00092-f002:**
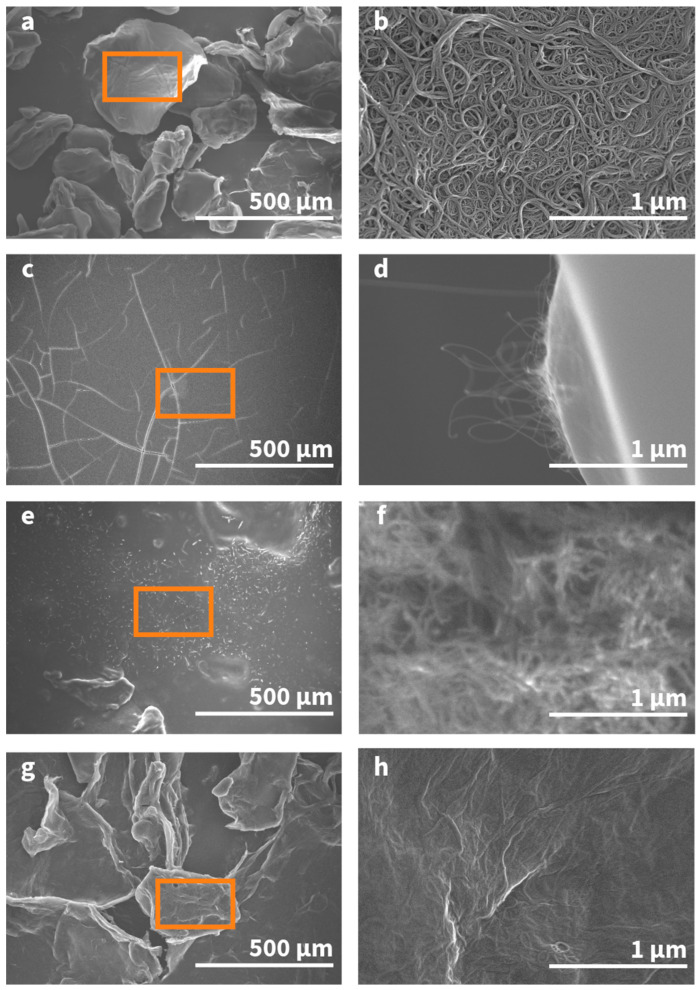
Scanning electron micrographs of SA-CNT dispersions for: (**a**) no surfactant at 100×; (**b**) no surfactant at 50,000×; (**c**) 15 *w*/*w*% GA at 100×; (**d**) 15 *w*/*w*% GA at 50,000×; (**e**) 8% Triton X-100 at 100×; (**f**) 8% Triton X-100 at 50,000×; (**g**) 1 mg/mL of GO at 100×; and (**h**) 1 mg/mL of GO at 50,000×. The rectangular box in the 100× magnification image represents the approximate image area in the 50,000× magnification image.

**Figure 3 nanomaterials-14-00092-f003:**
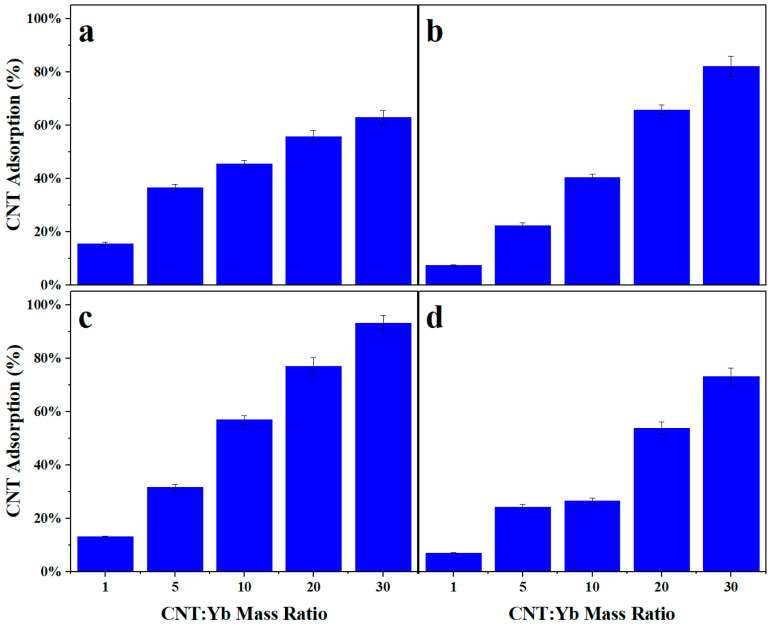
Percent adsorption of Yb onto SA-CNTs for all three surfactants tested: (**a**) 15 *w*/*w*% GA; (**b**) 8% Triton X-100; (**c**) 1 mg/mL of GO; and (**d**) no surfactant with increasing CNT to target mass ratios.

**Figure 4 nanomaterials-14-00092-f004:**
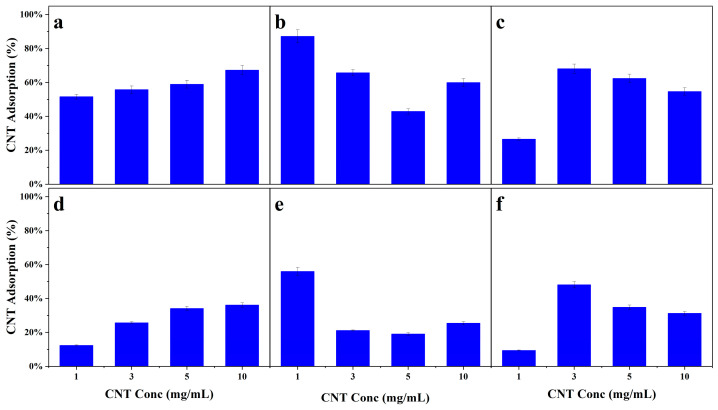
Percent adsorption of Yb onto CNTs at a 20 CNT to 1 Yb mass ratio for varying CNT concentrations in: (**a**) 15 *w*/*w*% GA w/SA-CNT; (**b**) 8% Triton X-100 w/SA-CNT; (**c**) 1 mg/mL of GO w/SA-CNT; (**d**) 15 *w*/*w*% GA w/CT-CNT; (**e**) 8% Triton X-100 w/CT-CNT; and (**f**) 1 mg/mL of GO w/CT-CNT.

**Figure 5 nanomaterials-14-00092-f005:**
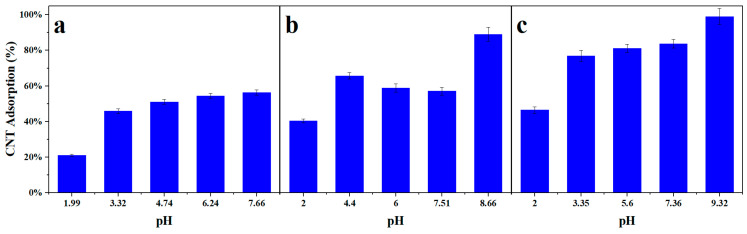
Percent adsorption on the SA-CNTs for varying pH of the surfactant at a 20 CNT to 1 Yb mass ratio in: (**a**) 15 *w*/*w*% GA; (**b**) 8% Triton X-100; and (**c**) 1 mg/mL of GO.

**Figure 6 nanomaterials-14-00092-f006:**
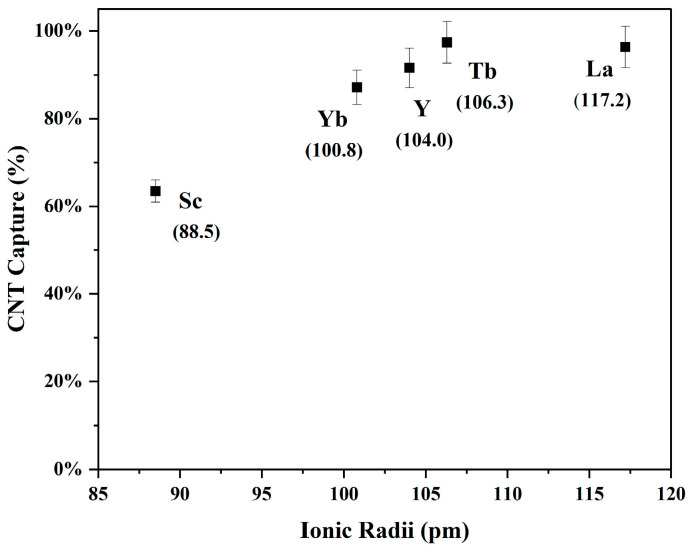
Percent adsorption of REEs onto SA-CNTs at a 20 CNT to 1 REE mass ratio in 1 mg/mL of GO at a pH value of 9.37.

**Table 1 nanomaterials-14-00092-t001:** Comparison of CNTs. Chemical Vapor Deposition (CVD); Catalyst (Cat.).

CT-CNTs	SA-CNTs
USD 250/g	USD 1350/g
D = 0.8–1.6 nm	D_Avg_ = 0.84 nm
L = 5–30 μm	L_Avg_ = 1 μm
Carbon Purity = 95 wt%	Carbon Purity = 95 wt%
Single/Double-Walled	Single-Walled
Functionalization = N/A	Functionalization = –COOH
Floating Cat. CVD Synthesis	Co-Mo Cat. CVD Synthesis

## Data Availability

Data are contained within the article and Appendix A.

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
