# Peer review of "Neutron Activation Analysis of Rare Earth Element Extraction from Solution through a Surfactant-Assisted Dispersion of Carbon Nanotubes"

_nanomaterials, 2023, doi:10.3390/nano14010092_

Round 1

Reviewer 1 Report

Comments and Suggestions for Authors

The paper entitled " Neutron Activation Analysis of Rare Earth Element Extraction  from Solution by a Surfactant-Assisted Dispersion of Carbon  Nanotubes. " focuses on the Rare Earth Element Extraction.  However, the authors can improvise this work farther. So my recommendation is to accept after major revision. The comments are attached:

1The manuscript needs to be improved, and the novelty of the study should be highlighted.

2Have you compared the results obtained using other materials for example?

3All the graphs have to be improved and keep the Figure format uniform.

Author Response

Have you compared the results obtained using other materials for example? 

-No, the exploration of adsorbents other than CNTs has yet to be explored. CNTs were chosen as the adsorbent, in this study for their unique characteristics (i.e., sizeable surface-area-to-volume ratio and hollow nano-sized structures).

All the graphs have to be improved and keep the Figure format uniform.

- The figures have been updated to be consistent and clear.

Reviewer 2 Report

Comments and Suggestions for Authors

1.      In the Title and text, the authors emphasize the use of neutron activation analysis to quantitatively determine the percent extraction of nuclides onto CNTs. Is there any other determination method besides neutron activation analysis? If other quantitative methods are available, it should be stated in the Introduction or text why the neutron activation analysis method was chosen. If this were the only quantitative method, there would be no need to emphasize it in the article title.

2.      When the authors analyzed the factors affecting the removal of nuclides by CNTs, it seems inappropriate to consider different CNTs manufacturers as one of the factors. According to the author's analysis (Line 245-288), the significant difference of CNTs provided by the two manufacturers (Sigma-Aldrich and Cheap Tubes) in the percent adsorption of nuclide is due to the different methods of manufacturing CNTs. Therefore, it is more reasonable to replace "different manufacturers" with "different process of manufacturing CNTs". There are too many manufacturers of CNTs, and if the CNTs manufactured by each manufacturer has different performance in adsorption of nuclides, then it is meaningless to discuss the factor of “CNTs manufacturer” in this article.

3.      All SEM images in Table S2 in the supporting information should be moved to the text, replacing he Fig.2 in Section 3.1. The information expressed by SEM images in Fig.2 is not clear enough.

4.      In Fig. 3, the percent adsorption of Yb by CNTs when they were added directly without the three surfactants should be provided.

5.      In Fig.3c, GO is a carbon material that can also remove Yb by adsorption. Compared with Fig.3a and 3b, is the high adsorption in Fig.3c the result of improved dispersion of CNTs or the adsorption contribution of GO to Yb?

6.      Line 287: Please give the full name of SWCNTs, when they first appear in the text.

7.      Line 284-288including Fig. S2):The authors suggested that MWCNTs had poorer adsorption performance for Yb than SWCNTs. How is this related to the difference in adsorption performance between CT-CNT and SA-CNT? It was mentioned in the article that CT-CNTs consist of a combination of single-walled and double-walled CNTs, but it did not clarify whether SA-CNTs were single-walled or multi-walled CNTs.

8.      Line 289-296: This part of the analysis cannot explain why the percent adsorption of Yb by CNTs in Fig.4a has been increasing with the increase of CNTs concentration.

9.      Line 385-387: Whether the adsorbed nuclide can be easily resolved from CNTs by dilute acid needs the support of experimental data, otherwise it cannot be written in the conclusion.

10.   The conclusions of the article need to be reorganized, and some conjectures cannot be put forward as conclusions. For example, the relationship between the adsorption of nuclides by CNTs and the electron density of CNTs, which needs to be proved by experimental data.

Author Response

1.) Although other analytical tools can be used to quantify REE extraction, NAA was chosen for its sensitivity towards trace metals in highly dense and opaque carbonaceous sample mixtures with minimal sample processing. This technique is also readily available for a minimal cost at UT-Austin.

2.) The manuscript has been revised to emphasize the the analysis of "different processes of manufacturing CNTs."

3.) SEM images have been moved from the Supplementary Info to the text.

4.) Percent adsorption of Yb onto the CNT adsorbent without a surfactant material has been included.

5.) When exploring the use of GO as a surfactant, the heightened capture is due to the low viscosity of the GO surfactant compared to Triton X-100. This lowered viscosity resulted in a higher mobility and subsequent higher capture of the Yb species.

6.) SWCNTs has now been defined in its first appearance. 

7.) The Sigma-Aldrich CNTs analyzed in this study were defined as being SWCNT in Table 1. It has now also been further clarified in the text.

8.) Due to the physical limitations of centrifuging and separating CNT solutions with concentrations greater than 10 mg/mL, these concentrations were not evaluated.

9.) Experimental results for the removal of nuclides from the CNT structure has now been included in the Supplementary Info.

10.) The wording in the conclusions has been updated with the commenters remarks. 

Reviewer 3 Report

Comments and Suggestions for Authors

This MS study two CNT in three media to remove REE. The data are self-consistent. It is possible to be published in Nanomaterials.

However, there are some confusions. It seems authors want to find cheap materials for REE removal. Then, CT-CNT is much cheaper than SA-CNT, and still have ca. 60 % efficiency (see fig 4). Logically, fig 5 and 6 should use CT-CNT.  

From fig 5, Triton X-100 at pH 8.66 has very similar efficiency to GO at 9.37, why authors only test GO for Sc, Y, Tb, La? Is there any reason? Also, what is the CNT concentration in Fig 6?

Therefore, the MS needs revise before acceptance.

Comments on the Quality of English Language

OK.

Author Response

  • This main focus of this study was to achieve the maximum extraction of REE from an aqueous solution regardless of cost-effectiveness.
  • Graphene oxide was chosen as the sole surfactant material in the ionic radii section due to its ease of handling. The lower viscosity allowed for simpler centrifugation and solution removal. (The CNT concentration is given in the text as 3 mg/mL) 

Round 2

Reviewer 2 Report

Comments and Suggestions for Authors

All my comments have been answered, and I agree to accept it.